# Effect of Mg Powder’s Particle Size on Structure and Mechanical Properties of Ti Foam Synthesized by Space Holder Technique

**DOI:** 10.3390/ma15248863

**Published:** 2022-12-12

**Authors:** Hongjie Luo, Jiahao Zhao, Hao Du, Wei Yin, Yang Qu

**Affiliations:** 1School of Metallurgy, Northeastern University, Shenyang 110819, China; 2Engineering Technology Research Center of Ministry of Education for Materials Advanced Preparation, Shenyang 110819, China; 3School of Naval Architecture and Ocean Engineering, Guangzhou Maritime University, Guangzhou 510725, China; 4Beijing Foton Daimler Automobile Co., Ltd., Beijing 101400, China

**Keywords:** titanium foam, magnesium space holder, porosity, percentage of open pore, yield strength, elastic modulus

## Abstract

Titanium foam has been the focus of special attention for its specific structure and potential applications in purification, catalyst substrate, heat exchanger, biomaterial, aerospace and naval industries. However, the liquid-state foaming techniques are difficult to use in fabricating Ti foam because of its high melting temperature and strong chemical reactivity with atmospheric gases. Here, the fabrication of Ti foams via the powder metallurgy route was carried out by utilizing both magnesium powders and magnesium particles as spacer holders, and Ti powders as matrix metal. The green compacts containing Ti powder, Mg powder and Mg particles were heated to a certain temperature to remove the magnesium and obtain the Ti foam. The results show that the porosities of the obtained Ti foam are about 35–65%, and Young’s modulus and yield strength are found to be in the ranges of 22–126 MPa and 0.063–1.18 GPa, respectively. It is found that the magnesium powders play a more important role than the magnesium particles in the deformation and the densification of the green compact during the pressing, and the pore structure of Ti foam depends on the amount and the size of the magnesium spacer holders after sintering.

## 1. Introduction

In the last twenty years, titanium and titanium alloy foams have drawn more and more attention as a futuristic material for their potential structural applications in aerospace, naval and biomaterial industries, as reviewed in [1,2,3] and references therein, combining the characteristics of both Ti and metal foams. Furthermore, the high melting point and excellent corrosion resistance of titanium and its alloys make them suitable for high temperature applications, such as catalyst substrates or heat exchangers [1,2,4].

The high melting temperature and strong chemical reactivity of titanium with atmospheric gases and with mold materials make solid-state foaming by powder metallurgy more promising than liquid-state foaming techniques. Powder metallurgy, on the other hand, can produce porous titanium parts and its alloys with more precise control of process variables and pore size [1]. Among the powder metallurgical production techniques for titanium and titanium alloy foams, the space holder method can yield porous parts with desirable size, shape, volume and distribution of pores. As using an appropriate space holder material, magnesium usually can be removed at low temperature without excessive contamination [3,5,6,7,8]. Of the available space holders for titanium and titanium alloy foams, including carbamide (urea) powders [7], ammonium hydrogen carbonate particles [9,10], polymer granules [11] and magnesium powders [3], different options have been used as space holders and more attempts have been made to synthesize Ti-foam using the sintering of Mg and Ti-powder compacts [3,12], followed by the removal of Mg, which is also called magnesium distillation, and regarded as an environmentally friendly method [13]. Three criteria were taken into account in choosing magnesium as the space holder in Ti and its alloy foam production. First, Mg solubility in Ti is expected to be negligible since the solid solubility of magnesium in titanium is very small, according to the binary phase diagram of titanium and magnesium [14], so that properties of Ti would not be affected. Second, Mg powders possess much higher strength than polymers, and so keep their size and the shape during the pressing treatment, which can reach and even exceed 500 MPa [15,16]. Finally, magnesium provides a reducing atmosphere that prevents the oxidation of Ti during sintering. In addition, in biomedical applications, magnesium can be removed by the body so that its incomplete removal from the foam does not constitute a drawback [12].

It is accepted that appropriate type, shape, size and amount of the spacer holders lead to a desirable pore structure [3]. An increase in pore size resulting from larger spacer particles has been reported [17]. A large amount of research has been carried out on the effects of the amount of space holder materials, such as ammonium hydrogen carbonate (NH_4_HCO_3_), on the powder metallurgy technique in terms of the properties of porous titanium and porous titanium alloys [18]. It was also reported that the porosity depends mainly on additive amount, while open porosity depends not only on the additive amount, but also the particle size of the additive [18]. Among the available applications, most significant efforts have been made to prepare open cell Ti-foam for its use as a functional material. In the trials that we conducted, there are two types of pores, namely macropores and micropores in the Ti foams manufactured by evaporation of magnesium particles from Ti/magnesium green compacts. Macropores with rough cell walls, produced by evaporation of large magnesium particles, were rather rounded and had a range of particle sizes. On the other hand, micropores in the cell walls and edges of the foams were those left between Ti powders due to incomplete sintering or removal of small magnesium particles, which connect the macropores. While the interconnections in low porosity samples were achieved only by those micropores, macropores were connected directly as well as through micropores in high porosity samples [19]. Thus, it is assumed that the size of Mg particles should be critical for open porosity, as it can control both pore size and connection in the obtained foams. However, to the knowledge of the authors, few reports are available about the effect of Mg particles’ size on pore structure, porosity and open porosity, namely the connection between pores, as well as some relevant mechanical properties for titanium foams, namely compression stress-strain profiles and elastic moduli.

Based on the assumption mentioned above, it is the purpose of this work to study the effect of the particle size of the Mg spacer on the foam structure, with a focus on open porosity and pore size, as well as also the effect on the mechanical properties of Ti foams using the powder metallurgical space holder technique. To achieve this goal, Mg was used as spacer for the fabrication of Ti foams. Some foams were made using small-sized Mg particles, referred to here as ‘powder’, and comparable to the Ti particle size. Other foams were made using Mg particles with a much bigger size range, referred to here as Mg ‘particles’. This paper, firstly, describes the structure of Ti foam samples, made by using the powder metallurgical space-holder technique, in which both magnesium powder (particle average size about 48 µm) and magnesium particles (in a size range of 270–1700 µm) were employed. The Mg particles were additionally sieved and the samples further divided into five groups according to the ranges of particle sizes: <270 µm, 270~380 µm, 380~550 µm, 550~830 µm and 830~1700 µm. Then, the micro-structure of the obtained samples was characterized, with special attention on the open porosity and pore size. The (compression) stress-strain curves were measured and the maximum stress and elastic moduli were defined. Finally, the compressive properties of the manufactured foams with varying degrees of porosities were investigated and compared. 

## 2. Materials and Methods

### 2.1. Fabrication of Ti Foam Samples

Production of Ti foams via powder metallurgy space holder technique was carried out by utilizing both magnesium powders and magnesium particles as spacers. Ti powders (98.9% purity, Beijing Xingrongyuan Co., Ltd., Beijing, China) with an average size of 26 μm and magnesium powders (99.5% purity, Yaoxie Metal Materials Co., Ltd., Xingtai, China) with an average of 48 μm were used in this work, as shown in Figure 1. The Ti powders exhibit irregular and polygonal shape with a general size of <50 μm, and the Mg powders are generally near olivary, with long diameter of <80 μm, which is bigger than the size of the Ti particles. The impurities in titanium powder are mainly aluminum, silicon and carbon, while the impurities in magnesium powder are mainly aluminum, silicon and zinc. However, the impurities in the two metal powders will not have much impact on preparation of titanium foam. Furthermore, magnesium particles (99.8% purity, Yaoxie Metal Materials Co., Ltd.) with a wide range of distribution (270–1700 μm) and an average particle size of 660 μm were also used as spacers. The Mg particles were near spherical in shape and almost all particles were similarly shaped. To evaluate the size effect of magnesium particles on the microstructure and properties of the Ti foams further, the magnesium particles were sieved and divided into the ranges of 830~1700 μm, 550~830 μm, 380~550 μm, 270~380 μm and <270 μm before mixing with the Ti powders, as shown in Figure 1c, where the Mg particle sizes are in the range of 830~1700 μm.

To investigate the size effect of Mg spacers on structure and mechanical properties of Ti foams, three groups of Ti foam samples were designed and fabricated using different amounts of the magnesium spacers. The Ti powders mixed with Mg powders with volume ratios of 50%, 60%, 70% and 80% were employed for the first group samples. Then, a mixture of Ti powders, Mg powders and Mg particles with a volume-ratio of 1:1:1 was employed for the second group samples. In this case, the Mg particles in the different size ranges were considered, respectively. Finally, considering the best results from the second group of samples, new samples were prepared by adding Mg powder to reach a total amount of 70 vol.% of Mg in the mixing. To be sure all the powders/particles mixed uniformly, the mixing procedures mentioned above were carried out as follows: ethanol solution (ethanol + octadecanol + stearic, acid + octadecyl amine, CH_3_CH_2_OH + CH_3_(CH_2_)_16_CH_2_OH + CH_3_(CH_2_)_16_COOH + CH_3_(CH_2_)_16_CH_2_NH_3_), which was necessary for the uniform coating of Mg powders/particles, especially the coarse Mg particles with the fine Ti powders and the improvement in formability, was added as a binder to the mixtures, and the titanium/magnesium mixtures containing different magnesium contents were mixed together for 90 min to ensure the homogeneous distribution of the titanium on the magnesium spacers; then, extra Mg powders for the proposed percentage were added into the mixture, which was further mixed for 30 min.

After uniform mixing, all the mixtures were heated at 60 °C for 30 min to evaporate the ethanol, then cold pressed at an subjected to a pressure of around 400 MPa for 20 min using a double-ended steel die to obtain green compacts with a size of 10 mm in height and 25 mm in diameter. Then, the compacts were slowly heated to 1150 °C and kept at that temperature for 4 h under high-purity argon gas for debinding, sintering and complete removal of magnesium. Finally, the sintered porous specimens were cooled down to room temperature. Each group had three parallel samples for subsequent microscopic analysis and performance testing when preparing titanium foam. The process flow chart of preparing Ti foam by space holder method is shown in Figure 2. It can be seen from Figure 2 that the magnesium particles are first completely wetted with the forming agent, and then mixed with the mixture of Ti powder and Mg powder to ensure that the mixture of Ti powder and Mg powder can cover the surface of magnesium particles evenly.

### 2.2. Structure Characterization of Ti Foam Samples

Porosities in the sintered specimens were determined by Archimedes’ principle, using a Sartorius precision balance equipped with a density determination kit. Fractions of open porosity were determined by weight measurements prior to and after dipping the samples in boiling paraffin at 170 °C for impregnation of open pores with liquid paraffin. Microscopic analysis was performed by scanning electron microscopy (SEM), using a Jeol 146 JSM 6400 microscope (JEOL Ltd. Akishima, Tokyo, Japan) equipped with the Noran System 6 X-ray microanalysis system (Thermo Fisher Scientific, Waltham, MA, USA). Both optical and SEM images on surfaces and cross sections were used to determine shape, size and distribution of the pores detectable by quantitative image analysis using a commercial software (KS400, Carl Zeiss Microscopy Deutschland GmbH, Oberkochen, Germany) after sintering of the specimens. After measuring the sizes of both green compact and the sintered porous Ti samples, the linear shrinkage was calculated by:(1)$$\Delta d_{G_{S}}=\frac{d_{G}-d_{S}}{d_{G}}\times 100\%$$
where *d_G_* is the height of the compressed green compact sample, and *d_S_* is the height of the sintered porous Ti sample, respectively.

### 2.3. Mechanical Properties of the Ti Samples

Mechanical properties of porous Ti samples were studied by the compression test on an INSTRON 55822 (INSTRON Cop., Norwood, MA, USA, 02062-2643) compression testing machine using the cylindrical shaped samples with a dimension of Φ20 mm × 8 mm. These samples were compressed at a strain rate of 0.5 mm/min at room temperature. Yield strengths and elastic modulus of the Ti foams were determined using 0.2%-off -set method and from curves fitted to the linear elastic regions of the stress–strain curves, respectively. Average compressive strength and modulus values were calculated from three specimens chosen randomly.

## 3. Results and Discussion

### 3.1. Microstructure

#### 3.1.1. Green Compact

Since the green compacts are composed of Ti powders and Mg powders/particles, the adhesion between the metal powders after mixing is beneficial to the formability of the preforms. It can be assumed that the adhesion between the metal powders mainly depends on the plastic deformation between the Ti powders and the Mg powders/particles. As the plasticity of titanium powder is lower, the magnesium powders/particles will be deformed in the pressing process and improve the preform shapeability. This assumption is confirmed by the pressed samples composed of the Ti powders, the Mg powders and the Mg particles (1:1:1, in vol.%), as shown in Figure 3. It can be found that both the Mg powders and Mg particles may deform from olivary or near spherical to flat after pressing in the green compacts, while the Ti powders keep the original shape. Especially for the Mg powders, the deformation of the Mg powders is inhomogeneous, which may stem from the nonuniformity of the stress distribution in the sample during pressing.

To determine the behavior of Mg powder and Mg particles in compression, the relationship between the content of magnesium and relative density of the green compact is given in Figure 4, where the size of Mg particles is fixed in the range of 830~1700 μm and the volume fraction of Mg particles is fixed at 30%. It was also found that the relative density of the green compact by the mixed Ti powders and the Mg powders, and the mixed Ti powders +30%Mg particles and the Mg powders, increases with the increasing volume fraction of magnesium powders; the relative density reaches 0.97 when 80 vol.% magnesium powders were mixed with the Ti powders only. The results confirm that the magnesium powders play more important role than the magnesium particles in the deformation and the densification of the green compact during the pressing. The magnesium particles are much bigger than the magnesium powders in size, which may lead to a shielding effect between the magnesium particles and titanium powders; thus, the relative density of the green compact by the mixed Ti powders + Mg particles + Mg powders is a little smaller than that of the mixed Ti powders and the Mg powders at the same Mg content. The shielding effect can be described as the schematic diagram shown in Figure 5.

It should be emphasized that the Ti powders with poor plasticity were separated by the Mg particles/powders, which is very helpful for buffering the stress between the Ti powders, improving the formability and increasing the relative density of the mixed powders. It is predicted that the pore structure and properties may be controlled by regulating proportion of the Mg particles/powders in the green compact, considering the application of the Ti foams, which will be discussed in the following part.

#### 3.1.2. Sintered Compact

The microstructure of the sintered mass at the cross-section normal of the compacted surface of Ti powders and Mg powders is shown in Figure 6. It can be seen that a number of micro-pores with a size of several tens of micrometers are present in the Ti foams, in which several small pores with a size no more than 10 μm also appear after sintering. The number of the micro-pores increases with the increasing content of the Mg powders from Figure 6a,d) It seems that the adhesion (or the adhesion area) of the Ti powders decreases with the increasing content of the Mg powders, and the small pores in the adhesion area also decrease or even disappear, especially when the content reaches 80%, which is attributed to the Mg lost through evaporation in the green compact.

Furthermore, the shrinkage of the green compacts with different volume fractions of magnesium powders after sintering was also evaluated along both the horizontal direction (perpendicular to the height of the cylindrical sample) and vertical direction (parallel to the height of the cylindrical sample); the results are presented in Figure 7. It is clear that the compacts shrink during sintering along both directions, probably due to the combined effect of the connection of pores and sintering shrinkage in the walls. The linear shrinkage increases proportionally with the increasing magnesium content in the green compact, and there is a critical magnesium content (volume fraction) in this work of around 62%, above which the compacts shrink more along the vertical direction than horizontal direction during sintering, which is attributed to the coalescence of the micropores during the sintering process and different deformation of the composite powders in the green compacts. It is accepted that the micropores will shrink, coalesce and disappear during the sintering; the larger the percentage of the micropores in the compact, the more significant the effect mentioned above will be, resulting in higher line shrinkage. On the other hand, the deformation of both Ti powders and Mg powders in the green compact is bigger along the vertical direction, resulting in more adhesion area between the Ti powders after pressing, which plays a more important role in the higher line shrinkage when the volume percentage of the Mg powder is bigger than 62%.

In the case of the green compacts by not only Ti powders and Mg powders but also Mg particles with volume ratio 1:1:1 after sintering, the obtained foams contain mainly two types of pores: macropores and micropores on cell walls, as shown in Figure 8. It can be seen from Figure 8 that with the increase in magnesium particle size, the number of macropores in Figure 8a–d decreases obviously, but the diameter increases. Meanwhile, the size of micropores on the pore wall is basically the same, and the number of micropores does not change obviously. It seems that both macropores and most of the micropores were produced by the removal of the spacers, as the shape and average size of the pores resemble to that of the spacers. In fact, the micropores should be attributed to the evaporated Mg powders, the nonmetallic impurities existing in the raw powders, the insufficient sintering and the volume shrinkages which occurred during the sintering process, while the macropores are obtained by the evaporation of Mg particles. Quantitative metallographic studies show that these micropores have an average pore size of 50 μm, while macropores are rather rounded and have an average size of 300 μm, 420 μm, 620 μm and 990 μm, although they can be as large as 1500 μm at some locations. In fact, the average size of the macropores and micropores depends on the sizes of the Mg powders and the Mg particles in the green compact, which are almost the same in value. It should be mentioned that the pore size is a little bigger than that of the size of the magnesium particles, which could be due to the fact that the magnesium particles evaporate by breaking the mechanical bonds with the matrix, which in turn creates larger sized voids, and the bigger the Mg particles are, the less the number of macropores. Furthermore, it is interesting to find that those macropores exhibit a feature of basically homogeneous distribution with less sharp corners, which contributes to decrease the stress concentration. Although the macropores in the samples with bigger Mg particles seem to be isolated from each other, they are in fact connected through the micropores on the cell walls, which will be discussed in the following part on open porosity. It is indicated that the structure of the obtained Ti foams depends on both the amount and the size of the magnesium spacers.

In this case, the shrinkage of the green compacts with different magnesium particles after sintering was also evaluated and is presented in Figure 9. It is indicated that the line shrinkage increases with the increasing size of the magnesium particles in the green compact, and the value along horizontal direction is bigger than that along vertical direction. Comparing with the results of the Ti foam with magnesium powders only in Figure 6, it seems that the Mg particles with a bigger size play a less important role in the shrinkage of the green compacts after sintering than the Mg powders, which is attributed to the decreasing average thickness of pore walls, and less shrinkage and coalescence of the micropores, resulting in the lower line shrinkage.

As seen in the results mentioned above, magnesium particles with sizes in the range of 550–830 μm were chosen in the following work, considering both controllability of pore structure and line shrinkage. When the volume fraction of the Mg particles and the Mg powders was fixed at 70 vol.%, the macropores per unit area decrease with the increasing fraction of Mg powders and their wall thickness increase from Figure 10a–e) with the increasing fraction of Mg powders, while the micropores per unit area on the pore walls increase, as shown in Figure 10. It is proven that the macropores in the sintered compacts stem from the evaporation of Mg particles, while micropores may mainly come from the evaporation of Mg powders. As with the samples of Ti powders and Mg powders, the line shrinkage of the green compacts increases with the increasing magnesium powder content and the value along vertical direction is almost the same as that along horizontal direction, as shown in Figure 11. This variation in line shrinkage is attributed to the isotropic of the big pores by magnesium particles and its more important role on the line shrinkage.

### 3.2. Porosity and Percentage of Open Pores

Porosity is a very important parameter of relative density and mechanical properties for porous metals, while the percentage of open pores, which is also called as open porosity, is an important criterion to evaluate the interconnectivity for applications, such as porous implants, to deliver and transport some fluids. In this work, all the specimens exhibit dependence of porosity on the amount of the Mg spacers, with open porosity in values over 82%. Less closed or isolated porosity measured in the samples are observed to vary non-systematically from sample to sample and probably stem from the voids already present in the powders/particles and formed during atomization process, as shown in Figure 6, Figure 8 and Figure 10.

Figure 12 presents change in total porosity and the percentage of open pores (open porosity) in the manufactured foams with volume fractions of magnesium powders in green compacts prior to foam production, in which open pores and closed pores constitute the total porosity of the manufactured foams. Moreover, the changes in open porosity of each foam sample are also given for comparison in the same figure. As expected, an increase in the volume fraction of magnesium powders in Ti/Mg powder mixture results in a higher (total) porosity for the Ti foams. However, the porosity in the foams does not match directly with the magnesium content initially added to green samples. It can be found that the porosity is about 20% less than the initial magnesium powder content, which may be attributed to excessive shrinkage resulting from the interconnection and partial combination of micropores during sintering. Above a critical magnesium content, around 70 vol.% for the present study, the gap decreases to 15%, indicating the decreasing shrinkage. As far as the classical accumulation theory, if the magnesium added is to 100% by volume, the porosity of Ti foams increases to a value around 74% porosity, which is well below the added magnesium content, implying the shrinkage and the coalesce of the micropores, while the content of micropores is reduced to zero as expected. On the other hand, if no magnesium were utilized during foam manufacturing, the maximum achievable porosity content would be 31.5%, which is residual porosity due to partial sintering of the Ti powders. Furthermore, the amount of open porosity increases with the increasing magnesium powder content, which reaches 98% when the magnesium powder content is more than 70 vol.%, indicating that the magnesium powder plays an important role in the interconnection between pores.

In the case of the green compacts with not only Ti powders and Mg powders but also Mg particles with volume ratio 1:1:1 after sintering, the relationship between the porosity, open porosity and the size of the magnesium particles is shown in Figure 13. It can be seen that both the porosity and the open porosity decrease with the increasing size of the magnesium particles, which is attributed to the variation of pore structure in the Ti foams. As the big pores stem from the magnesium particles, the number of big pores decreases and the distance between the big neighbor pores increases with the increasing size of magnesium particles, resulting in the higher shrinkage and poorer connectivity of the big pores; thus, both the total porosity and the open porosity decrease. However, it should be pointed out that the insignificant dependence of porosity on the size of magnesium particles in the green compact is seen in this work; the porosity decreases from 61.2% to 60.3%, while open porosity from 98.5% to 94.3%, when the size of the magnesium particles increases from 270~380 μm to 830~1700 μm. It is indicated that the size of magnesium particles determines the size of macropores in the sintered samples, but does not play an important role in the porosity and open porosity of the obtained Ti foams, which may be attributed to the poor shrinkage of the macropores during the sintering.

When the volume fraction of Mg particles (550–830 μm) and Mg powders added into the green compact is fixed at 70 vol.%, the porosity decreases with the increasing fraction of the Mg powders, while the open porosity increases, as shown in Figure 14. It can be found that the porosity decreases from 63.8% to 57.1%, while the open porosity increases from 91.3% to 99.1% when the volume fraction of the added magnesium powders increases from 25 vol.% to 45 vol.%. The decreasing porosity with the increasing volume fraction of magnesium powders is attributed the higher shrinkage from the higher number of micropores. It is proven once more that magnesium particles play a more important role in porosity, while magnesium powders play a more important role in open porosity (pore connectivity).

### 3.3. Mechanical Properties

In this section, the mechanical properties of titanium foam are further characterized by engineering stress-strain curves under compression conditions, and then the change characteristics of the curve are analyzed, as well as the factors affecting the yield strength and Young’s modulus.

#### 3.3.1. Adding Mg Powder Only

All the specimens exhibit typical compression stress–strain curves with varying volume fraction of the Mg powders in the green compact without Mg particles under compression loading, as shown in Figure 15. It can be seen from the curves that the maximum stress of titanium foam increases significantly with the decrease in magnesium powder addition, and the strain range covered by the maximum stress peak is wider, meanwhile, the stress fluctuation in this range is more obvious. Meanwhile, in the compression stress–strain curves, two distinct regions appear during deformation: a linear elastic region and a plateau stage in which the flow stress is nearly constant. In the present study, no densification stage was observed for the limited strain. Therefore, it is not that there is no existence of stress plateau zone when 50% volume fraction of magnesium powder is added, but that the stress plateau zone does not appear when the strain reaches 20% only. It is interesting to find that the stress-strain curves are smooth without any significant fluctuation, which indicates no stress concentration appears in compressing these samples. It has been proposed that cell edge bending is the dominant mechanism that controls the linear elasticity; the plateau is associated with the buckling and collapse of the cell walls [20]. Probably, strain localization initiates in the weakest part on the pore wall. Then the deformation propagates through the foam, resulting in a band of collapsed cells. However, in the cases of the samples by magnesium powders, the Ti foams possess homogeneous structure of pores without macropores, resulting in no stress concentration and no strain localization.

It can be also found from the compressive stress-strain curves of the Ti foams with different amounts of magnesium powders that the maximum stress decreases with the increasing content of magnesium powders. The value changes from a little more than 120 MPa in the case of 50 vol.% to only 20 MPa when the fraction increases to 80 vol.%, which is attributed to the increasing porosity. When the compressive stress reaches the maximum, brittle failure does not occur immediately, but deformation continues under low stress caused by the porous structure. The higher porosity leads to a lower stress in the deformation. It is accepted that the deformation mainly occurs through holes in the samples with open pore, while the deformation of the samples with closed pore mainly depends on the pore walls [20]. Thus, the strength and the elastic modulus of the samples decrease with the increasing open porosity at the same (total) porosity. On the other hand, in this study, it can be observed that the values of elastic modulus decrease with the increasing fraction of magnesium powders, which changes from 1.2 GPa to 0.06 GPa, as shown in Figure 16. The low modulus and its controllability of the porous titanium would prevent stress from shielding and produce a good matching with the mechanical properties of human bones, etc., which is helpful and promising to meet the mechanical requirements for biological applications.

#### 3.3.2. Adding Mg Powder and Mg Particle Together

In the case of the green compacts made including Ti and both Mg powder and Mg particles with volume ratio 1:1:1 after sintering, the relationship between the stress-strain and the size of the magnesium particles added into the compact is shown in Figure 17. As shown in Figure 16, the maximum stress of the curve is about 14 MPa when the granularity of magnesium particles is between 270~380 μm, 380~550 μm and 550~830 μm. However, when the particle size of magnesium reaches 830~1700 μm, the maximum stress decreases significantly. This shows that the larger change in particle size will also have a greater impact on the maximum stress. It can be found that the stress-strain curves are different from those curves measured on specimens manufactured with Mg powder only; apparent fluctuation appears in all curves, which is attributed to the local stress concentration during the compression and the local stress concentration comes from those macropores introduced by magnesium particles. It can be also found that the maximum stress decreases with the increasing size of magnesium particles. It should be mentioned that the value changes from about 14 MPa in the cases 270~380 μm, 380~550 μm, and 550~830 μm to only 8 MPa when the size of the magnesium particles increases to 830–1700 μm, although the corresponding porosity decreases from 61.2% to 60.3% only (Generally, the higher porosity leads to a lower stress in the deformation), which is attributed to the stress concentration by the bigger pores from the bigger size magnesium particles. During the compression, deformation appears generally at the weakest point in the porous structure. It is indicated that pore size or spacer size play a more important role on the maximum stress after it reaches 830 μm, although the strength and the elastic modulus of the samples increase with the decreasing porosity. On the other hand, in this study, it can be observed that the values of elastic modulus decrease with the increasing size of magnesium particles added into the green compact, which changes from 600 MPa to 240 MPa, as shown in Figure 18. It can be also found that the elastic modulus remains stable when the size of the magnesium particle reaches 550 μm, which is not clear at present.

When the volume fraction of the Mg particles (550–830 μm) and the Mg powders added to the green compact was fixed at 70 vol.%, their stress-strain curves are shown in Figure 19. It can be found the maximum stress (strength) increases from 10 MPa to about 20 MPa when the volume fraction of the added magnesium powders increases from 25 vol.% to 45 vol.%. Meanwhile, when the volume percentages of magnesium powders and magnesium particles were 25 vol.%:45 vol.%, 30 vol.%:40 vol.%, 35 vol.%:35 vol.%, 40 vol.:30 vol.% and 45 vol.%:25 vol.%, respectively, the corresponding elastic moduli of titanium foams were 409 MPa, 565 MPa, 230 MPa, 253 MPa and 1060 MPa, respectively. Although the ratio of magnesium powder and magnesium particles had a certain influence on the elastic modulus, it did not show an obvious regularity. The reason may be that the strain region of elastic deformation is very small, and the effect of the change in the ratio between magnesium powder and magnesium particles on elastic modulus is accidental to a certain degree. It is indicated that the magnesium powders in the mixed spacers for the green compacts play a less important role in the strength (mechanical properties) of the Ti foams. The lower strength is also attributed to the stress concentration by the bigger pores from the magnesium particles. 

It can be seen from the above discussion that Ti foams having porosities of about 35–65% have been produced by the use of magnesium spacers. In this porosity range, yield strength and Young’s modulus are found to be in the ranges of 22–126 MPa and 0.063–1.18 GPa, respectively, and the relative property changes with volume fraction of the Mg powders, the size of the Mg particles and the percentage of Mg powders in the mixed Mg spacers.

## 4. Conclusions 

Ti foams containing micropores in addition to macropores as a result of removing Mg spacers composed of both smaller powders and bigger particles by powder metallurgy are investigated in the present study. It is found that the magnesium powders play a more important role than the magnesium particles in the deformation and the densification of the green compact during the pressing. After sintering, the pore structure of the obtained Ti foams depends on the amount and the size of the magnesium spacers, and the bigger the Mg particles are, the smaller the macropores are, and the pore size is generally a little larger than that of the size of the magnesium particles due to the fact that the magnesium particles evaporate by breaking the mechanical bonds with the matrix, which in turn creates larger size voids. It is indicated that the magnesium particles determine the size of macropores in the sintered samples and porosity, while magnesium powders play more important role in content of open pores or interconnection between pores. It is proven that the magnesium powders in the mixed spacers for the green compacts play less important role in the strength (mechanical properties) of the Ti foams at a fixed content of Mg spacers. The lower strength is attributed to the stress concentration by the bigger pores from the magnesium particles. It is promising that the structure and mechanical properties of the sintered Ti foams can be modulated by varying the size and content of Mg spacer for green compact considering requirement of applications.

## Figures and Tables

**Figure 1 materials-15-08863-f001:**
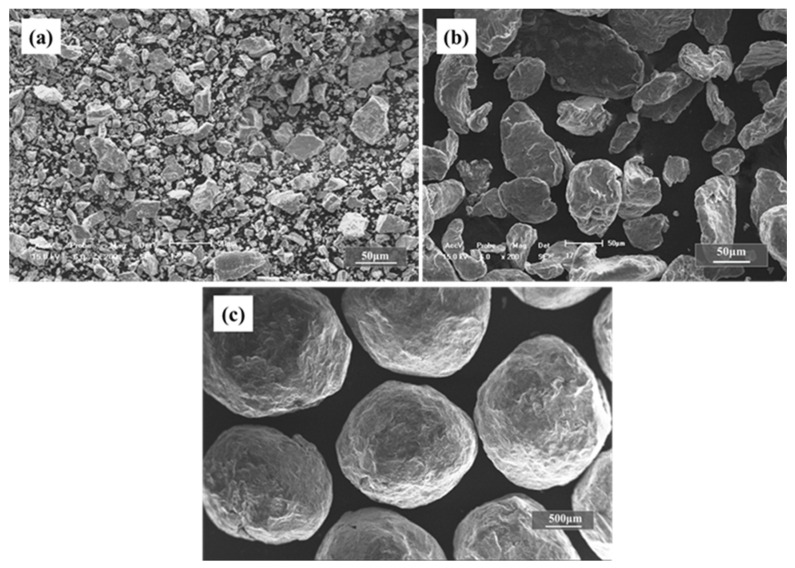
SEM images showed morphologies of Ti powders (**a**), Mg powders (**b**) and Mg particles (**c**).

**Figure 2 materials-15-08863-f002:**
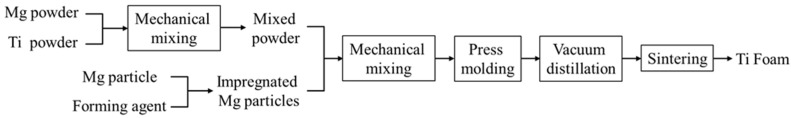
Process flow chart of preparing Ti foam by space holder method.

**Figure 3 materials-15-08863-f003:**
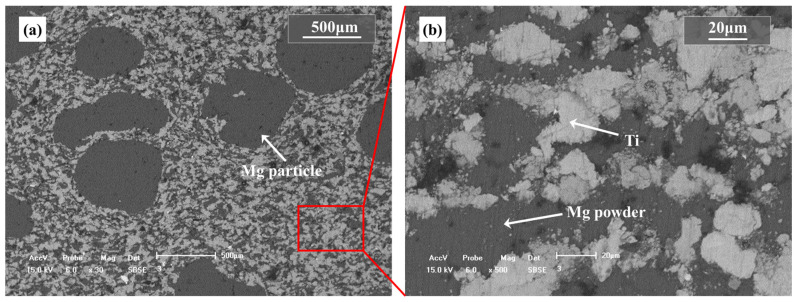
SEM images of prefabricated compact, (**a**) inner morphology on the cross section along pressing direction, (**b**) local amplification of (**a**).

**Figure 4 materials-15-08863-f004:**
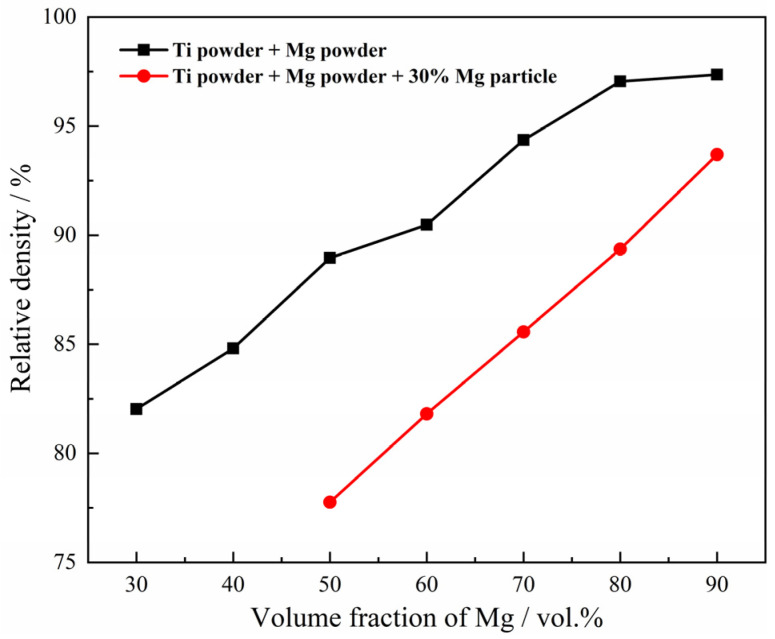
Relationship between content of magnesium and relative density of the green compact.

**Figure 5 materials-15-08863-f005:**
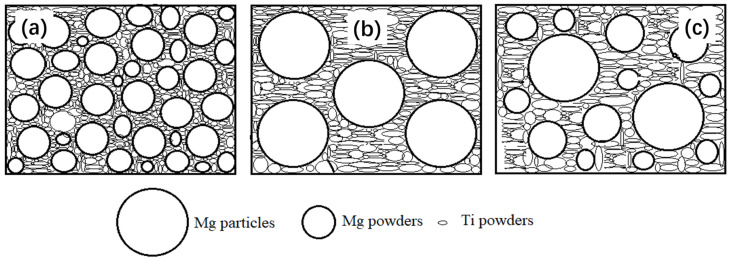
Schematic diagram of the mixed Ti powders, Mg powders and Mg particles in the green compact; (**a**) mixture of Ti powders and Mg powders, (**b**) mixture of Ti powders and Mg particles, (**c**) mixture of Ti powders, Mg powders and Mg particles.

**Figure 6 materials-15-08863-f006:**
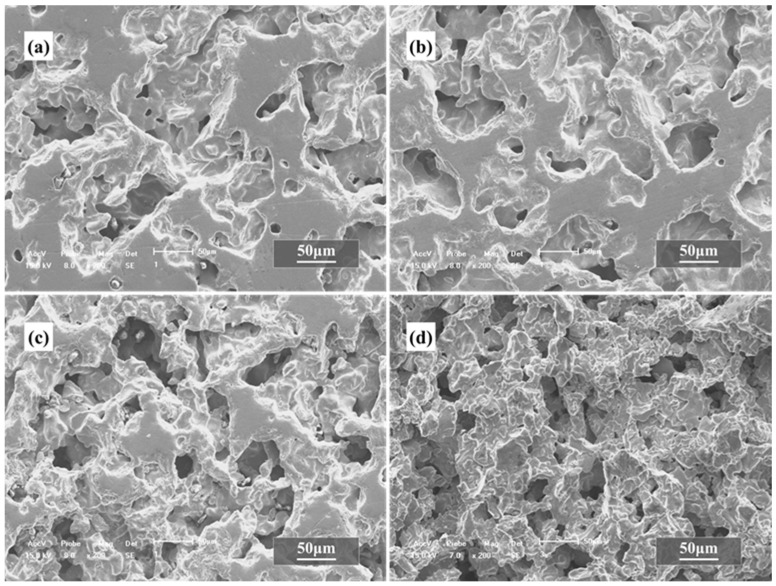
SEM images of porous titanium by mixed powders with different volume fraction of magnesium powders after sintering: (**a**) 50 vol.%, (**b**) 60 vol.%, (**c**) 70 vol.%, (**d**) 80 vol.%.

**Figure 7 materials-15-08863-f007:**
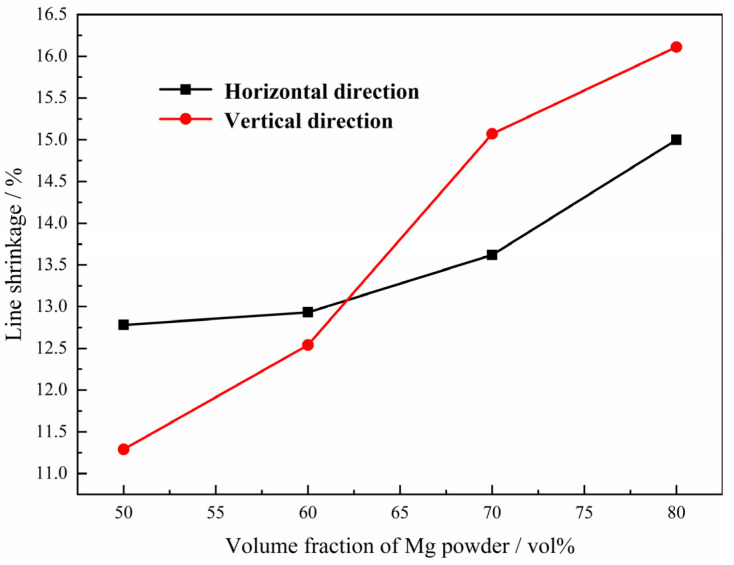
Relationship between line shrinkage and volume fraction of magnesium powder in the green compact after sintering.

**Figure 8 materials-15-08863-f008:**
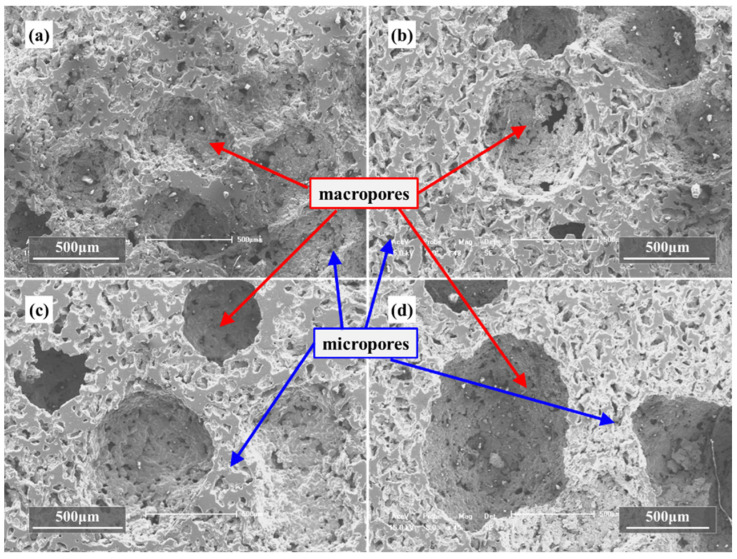
SEM images of porous titanium with different size of magnesium particles, (**a**) 270~380 μm, (**b**) 380~550 μm, (**c**) 550~830 μm, (**d**) 830~1700 μm.

**Figure 9 materials-15-08863-f009:**
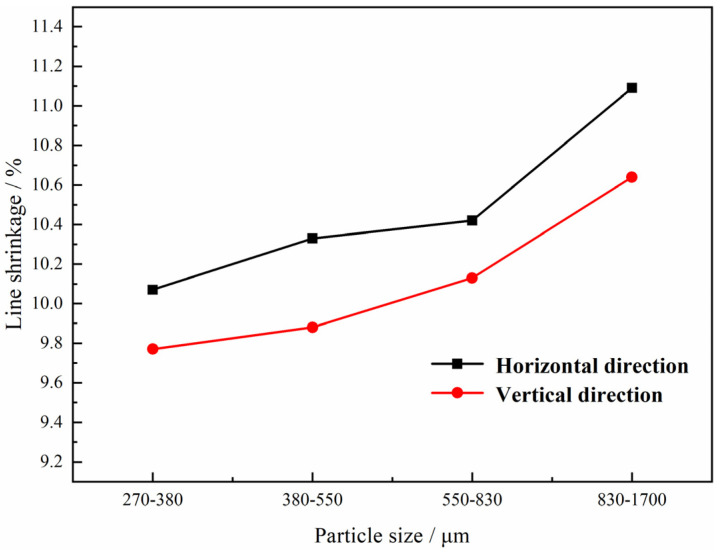
Relationship between line shrinkage and the size of magnesium particles.

**Figure 10 materials-15-08863-f010:**
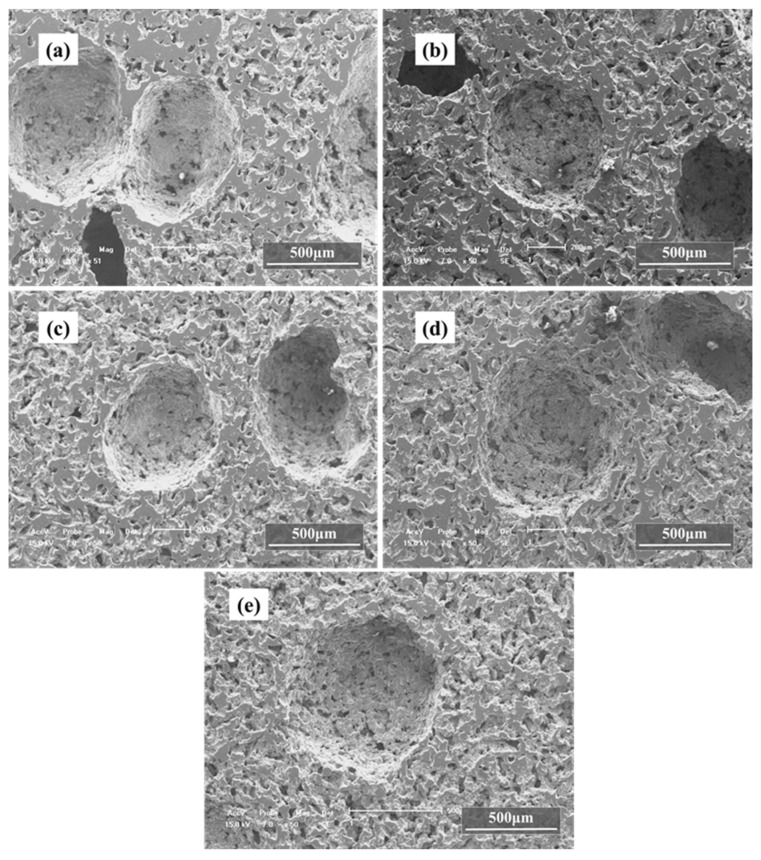
SEM images of porous titanium with 70 vol.% magnesium spacers after sintering, in which volume fractions of magnesium powders are 25% (**a**), 30% (**b**), 35% (**c**), 40% (**d**) and 45% (**e**).

**Figure 11 materials-15-08863-f011:**
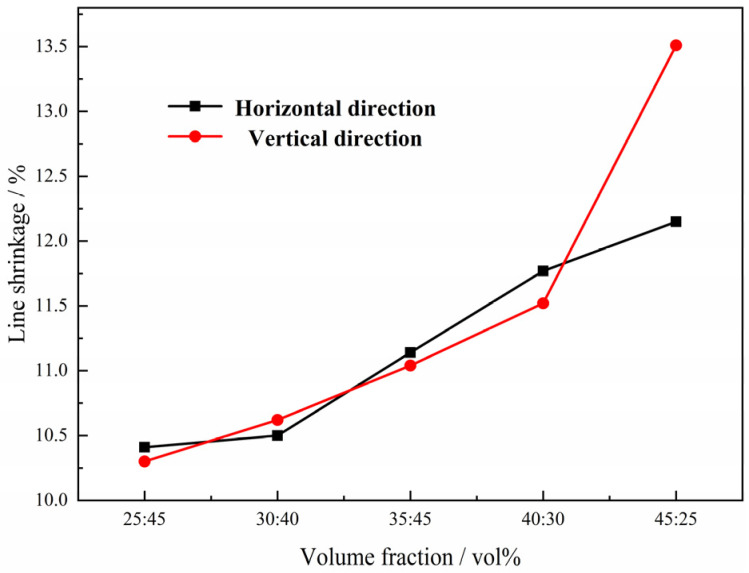
Relationship between line shrinkage and fraction of the magnesium powder when the volume fraction of the magnesium particles and the magnesium powders is fixed at 70 vol.%.

**Figure 12 materials-15-08863-f012:**
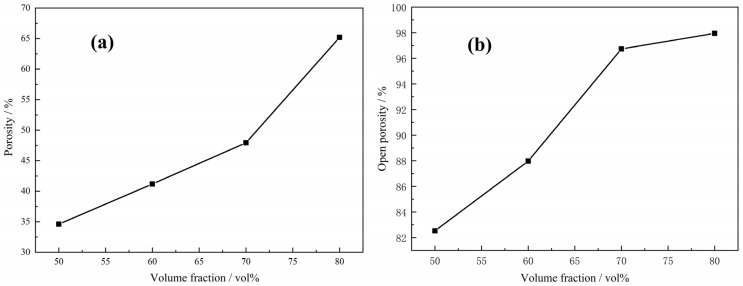
Relationship between porosity (**a**), open porosity (**b**) and volume fraction of magnesium powders in the green compacts by Mg powders and Ti powders.

**Figure 13 materials-15-08863-f013:**
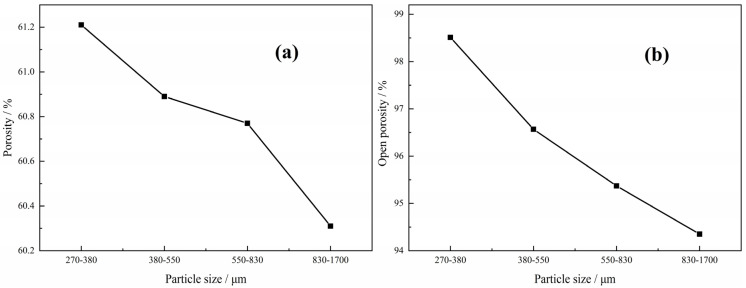
Relationship between porosity (**a**), open porosity (**b**) and the size of the magnesium particles.

**Figure 14 materials-15-08863-f014:**
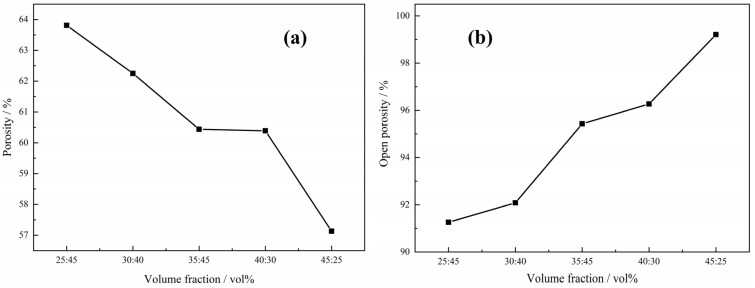
Relationship between porosity (**a**), open porosity (**b**) and the volume of the magnesium powders in the magnesium spacer added into the green compact.

**Figure 15 materials-15-08863-f015:**
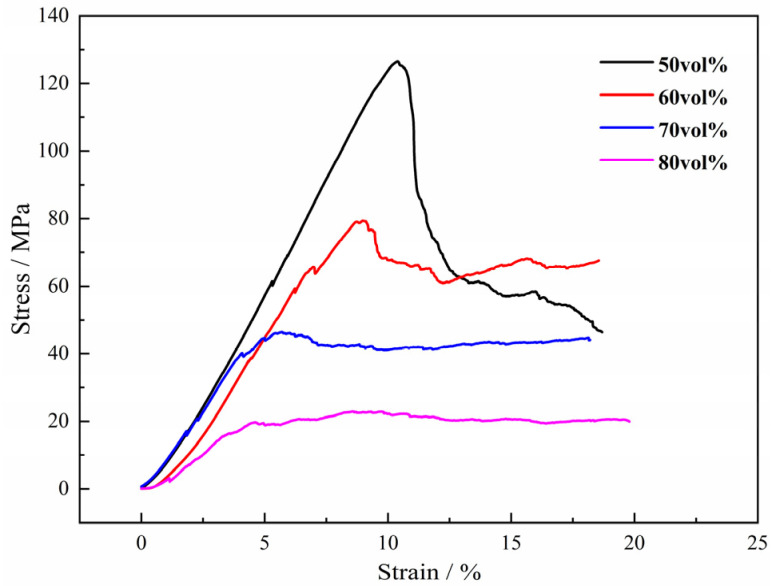
Curves of engineering strain-stress of porous titanium with different volume fractions of the magnesium powders.

**Figure 16 materials-15-08863-f016:**
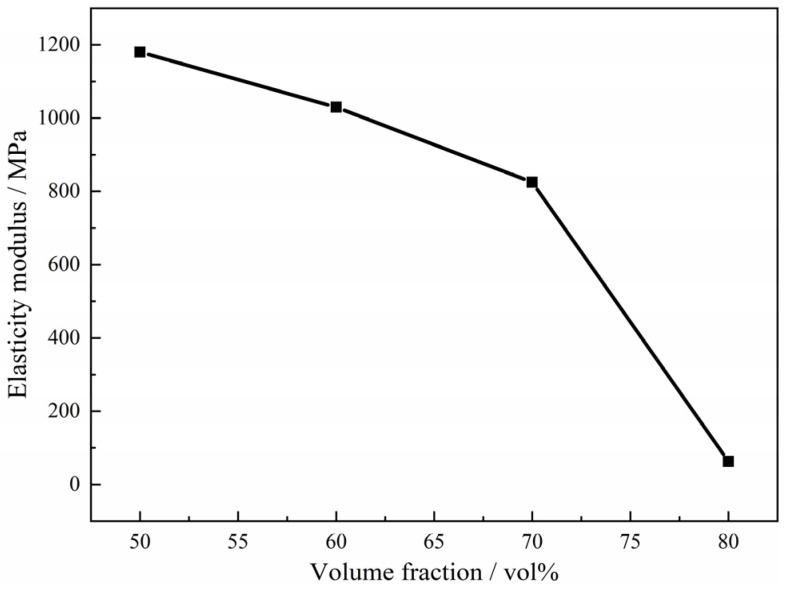
Effect of fraction volume of magnesium powder on elastic modulus of porous titanium.

**Figure 17 materials-15-08863-f017:**
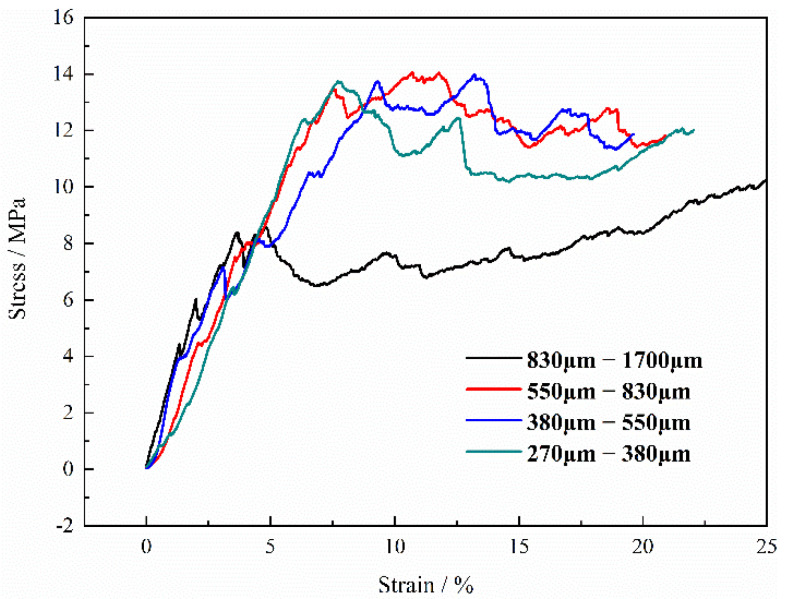
Curves of engineering strain-stress of porous titanium with different sizes of magnesium particles.

**Figure 18 materials-15-08863-f018:**
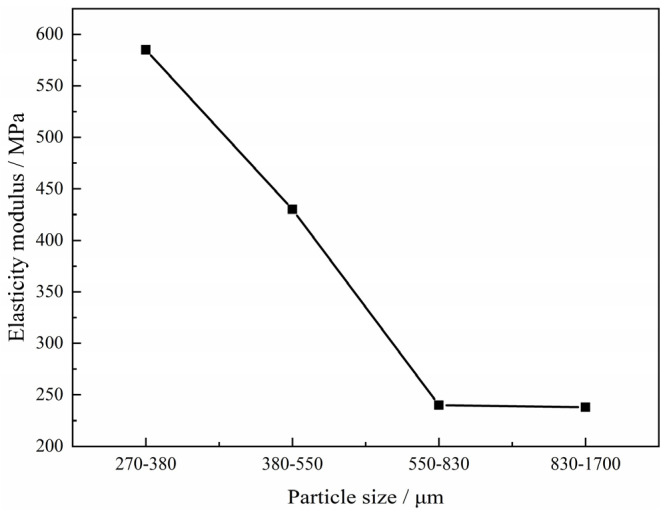
Effect of size of magnesium particle on elastic modulus of porous titanium.

**Figure 19 materials-15-08863-f019:**
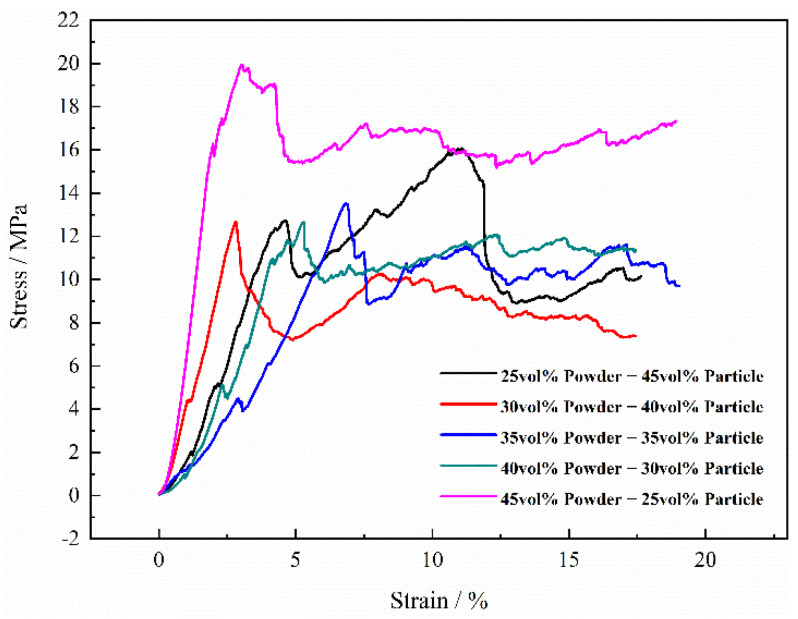
Curves of engineering strain-stress of porous titanium with different volume percentage of the magnesium powders and the magnesium particles.

## Data Availability

The data presented in this study are available in this manuscript.

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
