# Peer review of "Effect of Mg Powder’s Particle Size on Structure and Mechanical Properties of Ti Foam Synthesized by Space Holder Technique"

_materials, 2022, doi:10.3390/ma15248863_

Round 1
Reviewer 1 Report
1. There is no clear indication of the scale bars.
2. Adding more explanations or a schematic of the space holder method would be helpful.
3. Fig.4 should be labeled a, b, and c and described in the caption.
4. Please specify the micro and macro pores in Fig.7.
5. Are the stresses and strains true or engineering? Please modify the title of stress-strain curves.
6. For the 50vol%, no plateau is observed (Fig. 14). Why?
7. Why does the maximum stress decrease with the increasing size of magnesium particles (Fig.16)? Please explain in the text.
8. Why is the variation in the elastic modulus with volume fraction fluctuating (Fig.19)?
9. I think that it would be beneficial to compare these results with those obtained with Mg spacers fabricated from Ti foams.
Reviewer 2 Report
Reviewer Comment for Editor/Editor-in-Chief and/or Authors:
This manuscript provides a study on the size effect of Mg spacer on structure, pore size, and mechanical properties of Ti foams using powder metallurgical space holder technique.
This manuscript could potentially be suitable for publication, but it needs some minor revisions before it could be published.
1. Nothing mentioned about the reaction mechanism (from a physico-chemical point of view) between Mg spacer and Ti precursor. Much more explanation is needed with suitable citations to brush up this point.
2. The scale bar of all the SEM images (Fig. 1, 2, 5, 7, 9) is not clear at all, and must be rewritten to appear clearly on the images.
3. Figure 1 caption should be rewritten as “SEM images show the .........”.
4. Figure 2 caption, why you it "imagines"?
5. In Figure 2, EDX analysis should be done to support the metal's identification.
6. Too much Figures in the manuscript (i.e. 19), some of them should be moved to the SI.
7. References should be revised carefully as well as uniformly formatted. For example, please have a look to ref. 3, 5, 6, 7, 8, 9, 15, 16 and 18. these references have need to be revised compared to other references i.e. DOI and journal abbreviations.
8. One more point related to references, the references number is a bit low (i.e. 19) to support manuscript, therefore, it is highly recommended to include more recent citations to the manuscript.
Reviewer 3 Report
The paper
Size effect of Mg powders on structure and mechanical properties of Ti foam synthesized by space holder technique, by H. Luo et al., presents an interesting study of Ti foam made by sintering and based on a Ti-Mg powder mixing. The Authors present very detailed results of the study of the effects due to powder ratio and particle size of the Mg component used as space holder. The Authors present the results of the mechanical measurement of stress-strain done in the collection of different specimens analysed, what is a novel set of results related to Ti foams. The paper is well structured and presented, and correctly written, with a collection of quality results to support the discussions.
The interest in publishing these results is rather clear. However, before considering publishing, some points must be revised. I am confident that the Authors can check the comments and provide the (namely) missing information required to improve the draft.
- - - - REVIEW COMMENTS - - - -
(attached file)

Round 2
Reviewer 1 Report
It can be acceptable after responding to the reviewer's comments.
Author Response
We have revised the relevant content according to the second review comments. The specific revised contents were in blue font in the manuscript to distinguish historical revisions. Finally, we again thank all reviewers for their careful work!
Reviewer 3 Report
The Reviewer does appreciate the thoughtful review done by the Authors to provide an improved manuscript, including missing information and clarifying some details, both important to make a sound paper.
Few details are below, that could help (to the consideration of the Authors).
L11 Titanium foam has been paid more attention for its special...
The sentence seems not to express a full meaning. Check with the Editor or consider these alternatives, or alike:
Titanium foam has been the focus of special attention for its specific structure...
or
Increasing attention has been paid to titanium foam for its special...
L52 ... according to the binary phase diagram...
why no including the reference mentioned in the answer?
L87 ...as well as also THE effect
L135 ... shown in Fig.1 (c) where the Mg particle sizes are in the range of 830~1700μm
Check the figure1-c scale : 50 um? 500 um?
